# Influence of Anthropogenic Activities on Redox Regulation and Oxidative Stress Responses in Different Phyla of Animals in Coastal Water via Changing in Salinity

Abhipsa Bal [1,2], Falguni Panda [1,3], Samar Gourav Pati [1,3], Taslima Nasim Anwar [1], Kajari Das [4] and Biswaranjan Paital [1,*]

1   Redox Regulation Laboratory, Department of Zoology, College of Basic Science and Humanities, Odisha University of Agriculture and Technology, Bhubaneswar 751003, India
2   Zoology Section, Regional Institute of Education, Sachivalaya Marg, Bhubaneswar 751022, India
3   Department of Zoology, School of Life Sciences, Ravenshaw University, Cuttack 753003, India
4   Department of Biotechnology, College of Basic Science and Humanities, Odisha University of Agriculture and Technology, Bhubaneswar 751003, India
*   Correspondence: biswaranjanpaital@gmail.com; Tel.: +91-674-2397029; Fax: +91-0674-2397970

**Abstract:** Salinity is a decisive abiotic factor that modulates the physiology of aquatic organisms. Salinity itself is modulated by various factors—most notably by anthropogenic factors. In coastal regions, increasing salinity is observed mostly due to the elevated rate of evaporation under high temperatures, especially under global warming. In addition, many other anthropogenic factors, climatic factors, chemicals, etc., also contribute to the changes in salinity in coastal water. Some of these include rainfall, regional warming, precipitation, moisture, thermohaline circulation, gaseous pollutants, dissolved chemicals, wind flow, and biocrusts. Salinity has been found to regulate the osmotic balance and, thus, can directly or indirectly influence the biomarkers of oxidative stress (OS) in aquatic organisms. Imbalances in OS potentially affect the growth, production, and reproduction of organisms; therefore, they are being studied in organisms of economic or aquacultural importance. Salinity-modulated OS and redox regulation as a function of phylum are covered in this review. The literature from 1960 to 2021 indicates that the altered OS physiology under changing salinity or in combination with other (anthropogenic) factors is species-specific, even within a particular phylum. Thus, knowing the response mechanisms of such organisms to salinity may be useful for the management of specific aquatic animals or their habitats.

**Keywords:** anthropogenic activities; aquatic animals; antioxidant enzymes; oxidative stress; phylum-specific; redox regulation; salinity stress

## 1. Introduction

Salinity is one of the most important abiotic factors and is known to impart substantial effects on the physiology of aquatic animals. The variation in habitats' salinity is influenced by many anthropogenic factors in coastal regions [1]. Increasing salinity in coastal and other water bodies is notably modulated by the rate of evaporation under high temperatures and/or global warming [2]. Many other anthropogenic-induced climatic factors, chemicals, ions, etc., also modulate the salinity of coastal water; for example, variations in rainfall, regional warming, precipitation rates, thermohaline circulation, biocrusts, and pollutant loads—including gases, various dissolved substances, and chemicals/ions such as sulfate, sulfuric acid, nitrate, nitric acid, $Ca^{2+}$, $Mg^{2+}$, $SO_4^{2-}$, $HCO_3^-$, $Na^+$, and $Cl^-$. Exploring the source(s), mode(s), mechanism(s), and pattern(s) of the changing salinity in water bodies—especially in coastal belts—is always useful for maintenance of the habitat and management of the inhabitants. This is because the salinity of water significantly influences the physiology of its inhabitants [1–3]. Salinity is a major influential abiotic parameter in

maintaining the homeostasis of the inhabitants. About 97% of the water on the surface of the Earth is saline in nature. Such large saline water portions harbor millions of species in various ecological niches.

Salinity is measured in practical salinity units (PSU), g $L^{-1}$ or parts per thousand (ppt) (i.e., grams of dry weight of salt per kilogram of sea water), parts per million (ppm i.e., 0.0001% ($w/v$) salt in the water), etc. Salinity is the saltiness of water, so it depends on the concentration of salts in soluble form and, thus, is experimentally measured in terms of salt concentration. For example, it can be quantified and expressed in terms of the concentration chlorinity of seawater or the molality (i.e., moles of salts per kilogram of water) of sodium chloride solution, because both have the same vapor pressure [4,5]. The salinity of freshwater is usually found to be between 0.5 and 1 ppt, while it ranges from 0.5 to 30 ppt in brackish water and its value is >30 ppt in seawater. However, the salinity of some water bodies can be saturated up to 400 ppt (Figure 1). Large saline water bodies and fluctuations in their salinity ranges on the Earth's surface pose different impacts on the inhabitants, from growth and reproduction to aquaculture production. This encourages researchers to work on the different aspects of animals and plants with respect to changes in environmental salinity and their impacts on organisms [6,7].

The main factors that modulate the salinity in aquatic environments are the seasonal changes and the irregularities in the cycles of temperature and rainfall [6,8]. The concentration of salts in marine bodies increases as a result of thermal stress because it enhances the rate of evaporation. Anthropogenic activities and elevated sea levels, which limit the freshwater boundaries, can greatly regulate the rise in salinity. The changing salinity influences the physiology of every inhabitant of the aquatic ecosystem [9] (Table 1). This salinity fluctuation leads to imbalances in the health of the ecosystems which, consequently, affect the inhabitants' physiology, such as growth, reproduction, excretion, oxidative health, and genetics. Thus, these factors markedly disturb the distribution of species in saline water bodies. Euryhaline animals may withstand the fluctuation in salinity, while stenohaline animals must live in either hypo- or hypersaline regions [10,11]. In addition, salt uptake can influence the physiology of higher organisms in many ways [12–14]. In particular, the lower organisms are mostly affected by the changing salinity; hence, studies on salinity in environmental chemistry have gained more attention [15–18] (Figure 2).

Aquatic environmental factors significantly influence the overall physiology of the inhabitants [7]. The regulation of oxidative stress (OS) by the altered salinity is one of the environmental physiologies of major concern in animals [6,19–21] (Figure 2 and Table 1). The outcome as OS is initiated by the oxidation of lipids, proteins, or nucleic acids caused by excessive production of reactive oxygen species (ROS), and studies on OS physiology play a paramount role in environmental chemistry [22–27] (Figure 2). Animals consume oxygen through internal respiration for oxidative phosphorylation to generate adenosine triphosphate (ATP). This process is coupled with the electron transport chain (ETC). Electrons, while being transported via ETC enzyme complexes, have a probability of leakage at complexes I and III to reduce $O_2$ incompletely, producing ROS. The successful incomplete reduction of $O_2$ produces different ROS, such as superoxide radicals, hydroxyl radicals, and hydrogen peroxides ($H_2O_2$) during the first, second, and third reductions, respectively. Therefore, the ATP production and microsomal oxidation reactions produce ROS as toxic byproducts. Although minimal levels of ROS (especially $H_2O_2$) are required for several signal transduction processes, ROS are usually toxic when overproduced in cells and lead to the generation of OS [28], which is a problem that can affect the subsequent physiology of almost all living organisms [25].

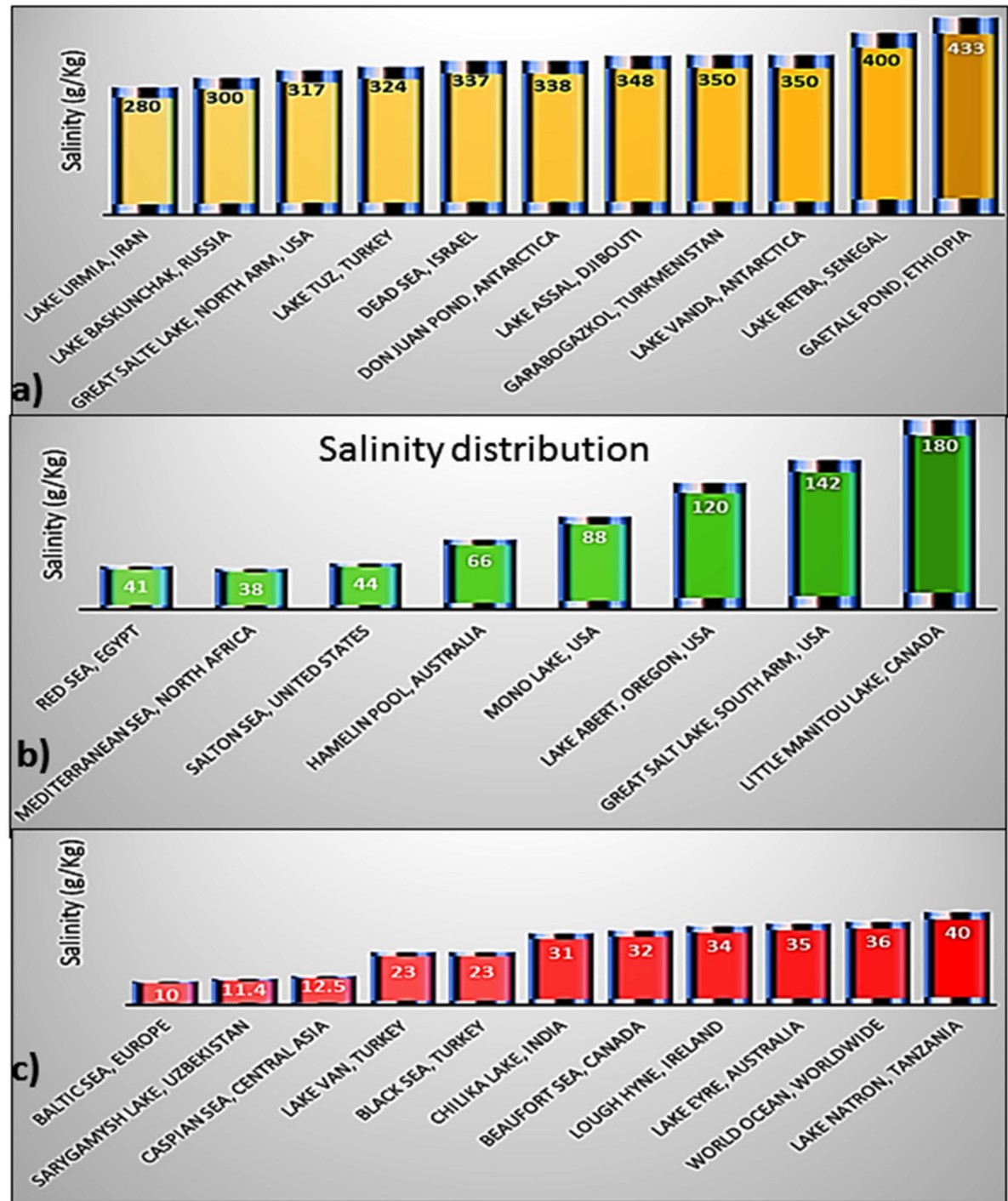

**Figure 1.** Distribution of various salinity regimes in different water bodies across the globe: Salinity (ppt) distribution in the ranges. (**a**) Salinity distribution in the range from 180 to 433 ppt, (**b**) Salinity distribution in the range from 38 to 142 ppt and (**c**) Salinity distribution in the range from 10–36 ppt.

**Table 1.** Low-salinity-modulated physiological responses (oxidative stress) in freshwater fish.

| Physiological/Biochemical Index | Salinity | Responses ↑ ↓ | Remark |
|---|---|---|---|
| Feed intake | −0.94 | ↓ ↓↓ | Increases metabolic depression |
| Gain in BW | −0.97 | ↓ ↓ ↓ | Modulate circulation, immunity, and blood functions such as $O_2^-$ and $CO_2$-carrying capacity. |
| Total length | −0.89 | ↓ ↓ ↓ | |
| RBC count | −0.96 | ↓ ↓ ↓ | |
| Hemoglobin | −0.99 | ↓ ↓ ↓ | |
| Thrombocytes | 0.38 | ↑ | |
| Lymphocytes | −0.80 | ↓↓↓ | |
| Monocytes | 0.38 | ↑ | |
| Eosinophils | −0.75 | ↓ ↓ | |
| Basophils | −0.72 | ↓ ↓ | |
| Neutrophils | 0.80 | ↑ ↑ ↑ | |
| LPx in the ARO | 0.93 | ↑ ↑ ↑ | Markers for tissue-level damage |
| LPx in the brain | 0.92 | ↑ ↑ ↑ | |
| LPx in the gills | 0.95 | ↑ ↑ ↑ | |
| LPx in the liver | 0.95 | ↑ ↑ ↑ | |
| LPx in the muscle | 0.95 | ↑↑↑ | |

Data were gathered on the response of a freshwater hardy fish (*Heteropneustes fossilis*) to low salinity (1 to 9 ppt) stress. The blood parameters were increased, perhaps to increase the rate of respiration and circulation efficiency under saline stress, whereas almost all tissues of the fish experienced oxidative stress in the form of lipid peroxide accumulation. The symbol ↑ represents an increase in the value under low salinity stress, while ↓ is used to indicate downregulation of the value. Single, double, or triple arrows are used when the correlation coefficient with the low salinity changes was found to be low, medium, or high, respectively. RBC: red blood cell; LPx: lipid peroxidation level, as a marker of stress; ARO: accessory respiratory organ.

The towering levels of ROS can be detoxified/removed through a series of reactions involving the antioxidant defense molecules present in the body. In the process, the generated superoxide radicals are counterpoised by the enzyme superoxide dismutase (SOD), which produces $H_2O_2$. Consequently, further reduction processes involve the action of the enzymes catalase (CAT) and glutathione peroxidase (GPx), which neutralize the toxic $H_2O_2$ and/or lipid hydroperoxides to water, respectively. Ascorbic acid (vitamin C), reduced glutathione (GSH), plant phenols, vitamin A, etc., are grouped under small non-protein antioxidant molecules that can potentially shield against the damaging ability of all active oxygen species or ROS directly and non-specifically. The harmful action of the oxidants is greatly counteracted by the glutaredoxins or thioltransferase systems. In addition, glutathione reductase (GR), GPx, and glutathione-S-transferase (GST) contribute as redox-regulatory helper enzymes that are conscripted for the removal of xenobiotics and recycle the conversion of GSH from oxidized glutathione. Lipid peroxides, protein carbonyls, and $H_2O_2$ serve as potential biomarkers for OS conditions and, in the process, when the activity of the antioxidant systems is low and cannot potentially neutralize the overproduced ROS and toxic molecules, cells experience OS. The glutaredoxin system also contributes to the redox regulation in living organisms. The above biomarkers of OS physiology in response to salinity are therefore considered to be important indices for comparative studies of several biological events [25,29] (Figure 2).

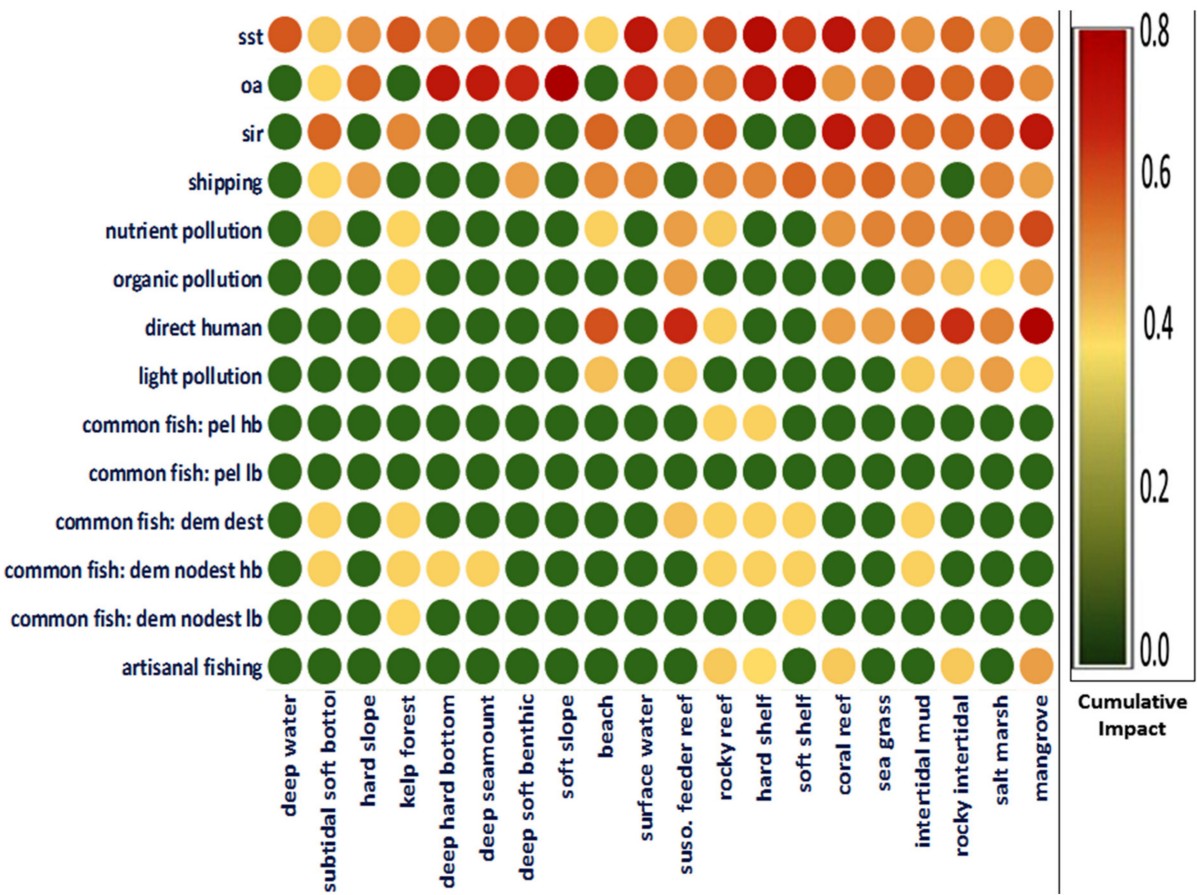

**Figure 2.** Influence of environmental salinity in different habitats. Salinity distribution and its cumulative impact on different aquatic animals—especially on fish and fisheries—are found to be highly relevant. Different zones, such as deep sea or mangrove habitats, have specific fish and fisheries values that are highly influenced by the salinity regime.

In the current review, we performed a systematic survey of the literature and reviewed the importance of OS in different phyla in relation to salinity changes. Information was gathered to reveal the associations between OS and altered salinity as a function of phylum. Studies indicate that the OS responses of animals are phylum-specific, or even species-specific within the same phylum [6,10,30–47]. However, reviews on this important avenue of salinity based on "OS physiology in animals belonging to different phyla" covering both osmoregulators and osmoconformers have not been covered. Very few reviews have reported the effects of osmotic challenges in aquatic animals, and mostly in invertebrates [7,48]. We believe that this is the first review on salinity-induced OS responses in different phyla of aquatic animals. In this article, we review the above aspects, covering OS and redox regulation in both invertebrate and vertebrate models under changes in salinity as a function of phylum. Modulation of salinity—especially in coastal belts—is also covered in this review. The review of the literature was strictly focused on peer-reviewed articles that were collected from databases such as PubMed, Scopus, Google Scholar, Science Direct, etc. The references used in this paper were organized and filtered with the help of Medley software.

## 2. Influence of Anthropogenic Activities and Climatic Factors on the Salinity of Coastal Water

Salinity, among several abiotic factors—such as turbidity, dissolved oxygen, temperature, pH, total dissolved solids, etc.—contributes to the dynamics of coastal aquatic habitats and their biodiversity. The biological communities thriving in coastal aquatic ecosystems are progressively influenced by optimal levels of salinity. The molecular, morphological,

physiological, biochemical, and ecological prospects of an organism are sustained by an optimal level of salinity in the water in which the organism survives. The osmotic concentration of an aquatic environment facilitates the distribution of the species in coastal habitats with respect to several auxiliary factors. These include the patterns of feeding and nesting, body fluid regulation, cellular homeostasis, regulation of osmotic genes, etc. [49–51]. The differential patterns of salinity shifts and the water levels recorded in the last few decades have led to devastating consequences for coastal regions. These regions experience erosion of the natural coastline, loss of biodiversity, contamination of freshwater biota due to saltwater invasion, etc. Further shifts in salinity levels in those coastal areas can disproportionately intensify the ramifications for the coastal ecosystems. In the present scenario, these coastal areas frequently experience variable salinity levels in absolute terms, mostly due to global climatic disturbances and anthropogenic events [52–54].

Today, the temperature patterns of the Earth are altered significantly and are expected to fluctuate even further over time under the influence of climatic changes induced by anthropogenic activities [55]. Such fluctuation in climate is estimated to affect the physico-chemical parameters of water—especially the water levels in the ocean—and, thus, to affect the levels of water salinity [56,57]. The ocean contains nearly 97% of the total water and is responsible for 86% of the Earth's total evaporation [58]. Therefore, various national and international programs are undertaken to monitor the fluctuations of water parameters, including salinity. According to the Intergovernmental Panel on Climate Change (IPCC), climatic changes such as increased global warming due to anthropogenic activity will affect organisms residing mostly in semi-arid and Mediterranean regions, due to a 25–30% increase in the rate of evaporation [59–61].

Higher salinity levels are observed where the rate of evaporation is high [62,63]. Similarly, lower salinity levels are observed where there is a high rate of precipitation and mixing of runoff water with the ocean [64,65]. The range of salinity varies from 34 to 36 ppt across the oceans, with some exceptions [66]. There are also some places where rainfall is limited but the warm wind evaporates the water and, thus, reduces the water level, ultimately resulting in increased salinity [67]. The Atlantic Ocean is one of the examples where such a case is observed. Similarly, a very high rate of evaporation is observed in the Mediterranean Sea, resulting in a salinity level of about 38 ppt [68]. Therefore, an increase in global warming can enhance the salinity levels of the Atlantic Ocean and Mediterranean Sea.

Some oceans have very low salinity, where there is heavy rain and the mixing of water from freshwater areas takes place. The coastal zones in the Antarctic and the Arctic can be affected by strong decreases in salinity due to ice melting and runoff [69], whereas the salinity levels of offshore Antarctic and Arctic waters have been observed to be 34 and 30 ppt, respectively. Thus, the increase in the Earth's surface temperature due to various anthropogenic activities can melt glaciers and increase water levels. This ultimately reduces the salinity levels. The Baltic Sea in Northern Europe has low-salinity regimes accounting for ~10 ppt due to heavy mixing of freshwater into the sea [66]. Therefore, increased global warming can increase the evaporation rate and, subsequently, could increase the salinity levels in different water bodies [7].

A 2 ppt increase in salinity can increase the physicochemical characteristics of water, such as density and buoyancy [66,70]. The higher salinity of water can cause denser water to sink below less dense water, which can cause convective mixing of two water levels. The increase in salinity has an even greater effect at low temperatures, because the combination of these two factors causes the ocean water to increase its density; as a result, it could sink lower—especially towards the bottom [58]. Conversely, a decrease in the salinity of the water can increase the stratification effect and, thus, prevent the mixing of waters [71]. The elevated salinity under the high rate of evaporation causes the formation of salt fingers that can imbalance the equal distribution of salts in water. Similarly, low salinity causes a barrier layer that could can vertical exchanges between two layers of the water [58].

The unprecedented rate of anthropogenic activities has put substantial pressure on aquatic environments in general, and on the species residing within them in particular. Anthropogenic inputs may be categorized into hazardous solid wastes, gaseous emissions, and contaminated liquid discharge. Heavy metal contamination of water bodies due to mining, the release of industrial and household sewage discharge due to anthropization, agrochemical-led agricultural practices, poisonous gaseous emissions from industry, introduction of xenobiotic compounds, the use of antibiotics, etc., have significantly contributed to fluctuations in water quality parameters, leading to a stressful aquatic environment—especially in coastal regions. Furthermore, these anthropogenic events are direct or indirect causative factors of physicochemical changes in the coastal regions, thereby driving the shifts in salinity. The change in salinity levels subsequently influences the aquatic health of the coastal and marine ecosystems in multiple ways, e.g., resulting in differences in plasma concentrations, improper regulation of body fluids, variable cellular structure, ROS formation, generation of OS, etc. [72–74]. Despite the fluctuations in salinity levels, several investigations have reported that some species have become resilient to hypo/hypersaline environments to a certain extent and play a major role in biomonitoring [7,75]. The uncontrolled exponential increase in human populations and their related activities have been found to induce climatic changes globally.

Incessant anthropogenic activities of imprudent nature have been seen to be counterproductive for global climate change. The anthropogenic stimuli and natural factors have individually contributed to the dimensional shifting of ecosystems along coastal regions in every possible way. The intensive intervention of human activities has taken the edge off the long-term climatic disturbances. However, the anthropogenic vectors co-interact with climatic changes and exacerbate the alterations of abiotic factors in the coastal hydrological ecosystems by increasing or decreasing the rise in sea levels, the concentration of sediments, and fluctuating salinity levels. Global climatic changes are characterized by global increases in temperature, reduced rainfall, frequent cyclonic storms and droughts in the coastal regions, El Niño–Southern Oscillation, etc. [76,77]. Frequent drought-like conditions due to the irregular distribution of rainfall have led to interruptions of salinity, thereby causing salinization of the hydrological ecosystems along the coastal regions. The coastal regions are vulnerable to cyclonic storms and tsunamis, which gradually lead to an increase in the salinity of the groundwater aquifers and the freshwater ecosystems in the close vicinity of coastal regions [54,78,79]. Additionally, the climate-driven salinity instabilities have imposed severe detriments on the population and dynamics of biological communities. Hence, the above natural disturbances cause the loss of biodiversity, along with an acute impact on the socioeconomic conditions of human populations in the coastal areas.

The coastal regimes are home to species belonging to different taxa that are of ecological importance. Fluctuations in salinity in these regions have induced shifts in diverse species communities at the expense of higher energy expenditure, stimulating the stress physiology of the species by reducing their overall efficiency. These changes not only threaten the diverse species communities but also boost the biological invasion of new species and, subsequently, cause a decline in their ecological significance. Additionally, the livelihood of the humans inhabiting these areas is comprehensively affected due to climatic disturbances and/or anthropogenic inputs. Therefore, maintaining an optimal level of salinity for sustaining life and balancing the equilibrium—especially among the aquatic communities—is a constant need for ecologists, conservation biologists, and environmentalists.

## 3. Salinity and Responses of Animals from Different Phyla

### 3.1. Physiology of Animals under Changing Salinity

Loss of biodiversity is observed in many wetlands due to salinization of water bodies, as the concentrations of salts and other inorganic ions increase for several reasons. Some of these causes include the breaking and silting out of the catchment areas of rivers and

water bodies. The other processes that can lead to increases in salinity are o water cycles such as rain, evaporation, sea spray, glacial/interglacial cycles, etc. Adding to the above possible reasons, salinization can be caused by anthropogenic activities such as an influx of town/urban water drainage, demolition of foliage, exhaustive irrigation on land, regulation of rivers, de-icing, salting out, chemical processes, global warming that results in rising sea levels, or excessive mining works. Huge reductions in biodiversity indices that lead to unhealthy ecosystems can affect many aquatic organisms—especially ectotherms—and for this reason a lot of work on various aspects of various organisms has gradually been conducted from the 1960s to the year 2021 [7,80–83] (Tables 1 and 2; Figure 3).

**Table 2.** Cellular energetic responses of different aquatic animals under salinity stress.

| Kinases in Signaling Pathways | Species in Which the Kinase Response is Observed |
| --- | --- |
| Mitogen-activated protein kinase | Tilapia (*Oreochromis mossambicus*) |
| | Killifish (*Fundulus heteroclitus*) |
| | Turbot (*Scophthalmus maximus*) |
| Myosin light-chain kinase | Japanese eel (*Anguilla japonica*) |
| Focal adhesion kinase | Killifish (*Fundulus heteroclitus*) |
| Osmotic stress transcription factor 1 | Mozambique tilapia (*Oreochromis mossambicus*) |
| | Blackhead seabream (*Acanthopagrus schlegelii)* |
| | Japanese eel (*Anguilla japonica*) |
| | Mozambique tilapia (*Oreochromis mossambicus*) |
| | Nile tilapia (*Oreochromis niloticus*) |
| | Medaka, the Japanese rice fish (*Oryzias latipes*) |
| | Zebra fish (*Danio rerio*) |

Any type of stress generally induces energy metabolism. In the ratio between energy demand and supply, whether remaining constant or increased, the former always has a higher value under stress in general and under salinity stress in particular. High salinity and low salinity cause osmoconformers and osmoregulators, or saline or freshwater animals, respectively, to need extra energy, for which different kinases are activated to meet the demand, as mentioned in the table. Modified based on the work of Bal et al. [7].

Salinization influences the life cycles of many animals. Activities such as the reproduction, spawning, growth, and migration of aquatic organisms are intensively influenced by salinity. For example, the micronutrient contents and gene regulation in stock broods of Atlantic salmon were linked to salinity. The growth of the fish is also influenced by seasonally induced salinity changes [84]. However, the influence of changing environmental salinity on the OS physiology in organisms has been scarcely addressed in the literature. The OS physiology begins with the consumption of $O_2$ by the organisms, and the mitochondria act as the main hub for the generation of ROS. The effects of salinity on mitochondrial behavior have been studied by a few researchers [19,20] (Figure 4). It is worth mentioning that energy catabolism by kinases is very distinct in animals under varying levels of salinity [85–95]. Therefore, this could be instrumental in igniting many associated cellular pathways in animals to respond in a variety of ways to the changing salinity (Tables 1 and 2). For example, mitogen-activated protein kinase action was observed in tilapia (*Oreochromis mossambicus*) and killifish (*Fundulus heteroclitus*) (Table 2).

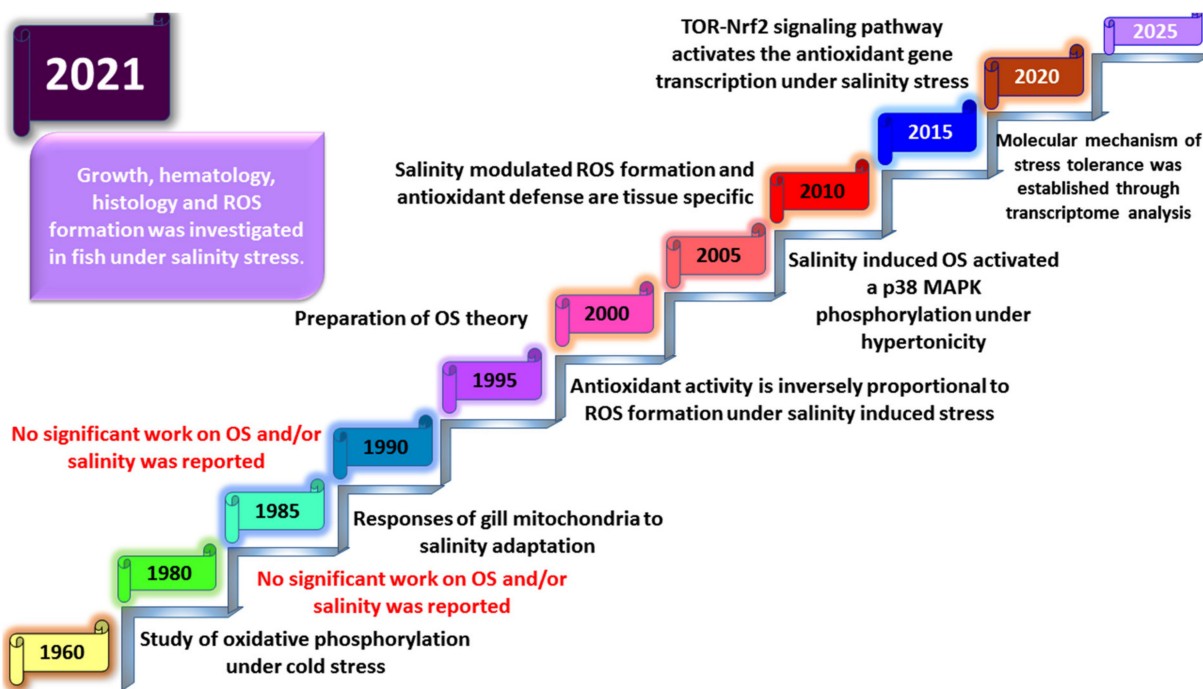

**Figure 3.** Works in the last 60 years on different aspects and effects of salinity in organisms: Literature from PubMed with specific terminologies was examined to collect overall information on the work types. The results indicate that work on salinity stress in animals has begun to understand the influence of salinity on overall physiology (e.g., growth, morphology, development, etc.) and has reached the target level of rapamycin and nuclear factor erythroid 2-related factor 2 expression, which regulates the expression of the redox-regulatory enzymes or for redox sensing.

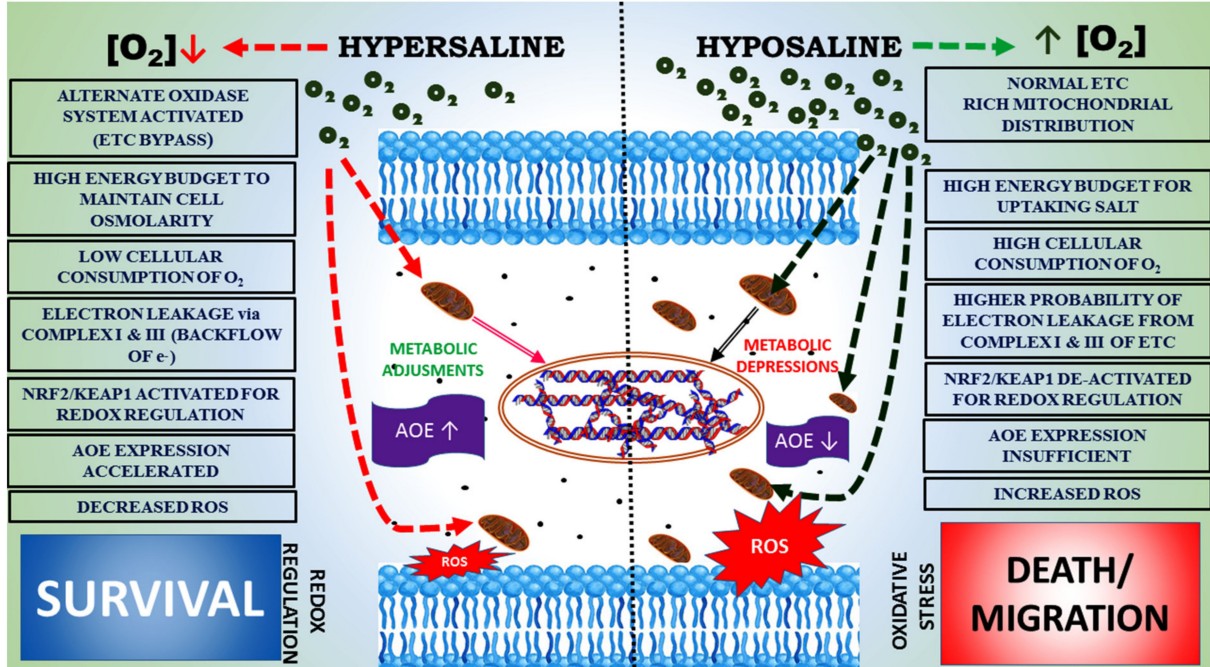

**Figure 4.** Responses of mitochondria to the changing salinity in relation to hypoxia or hyperoxia: Hyposalinity with hyperoxia can modulate mitochondrial respiration, which can lead to the accumulation

of reactive oxygen species following normal oxidative stress pathways, i.e., high rates of respiration lead to high rates of reactive oxygen species generation due to electron leakage via complex I and III enzymes. The normal electron leakage and lipid radical production in the chain may lead to more accumulation of reactive oxygen species, which can only be controlled by or recovered under high redox adaptation or by shifting the animals to isosmotic environments. Under special conditions, mitochondria can activate alternate oxidase systems to reduce electron leakage and reactive oxygen species generation. Associated signaling or metabolic pathways, as shown in the figure, are found to be regulated in cells under hypo- or hypersaline conditions. The photos of the species were collected from Google Images with a creative common attribution license.

Mitochondria act as the hub for the generation of active oxygen molecules [29], and the changes in the mitochondrial respiration and their respiratory chain enzymes are substantially modified by the changing salinity in the environment of aquatic organisms [6,19]. The osmoregulatory mechanisms induced by the changing salinity are associated with the respiratory behavior of mitochondria and are observed to be species-specific under hypo- and hypersaline conditions [19,20] (Figure 4). Different phyla can behave differently in response to the influence of the changing salinity on their blood or hemolymph osmotic pressure, which is influenced by environmental osmotic pressure. In turn, their OS responses vary under different salinity conditions (Figure 5).

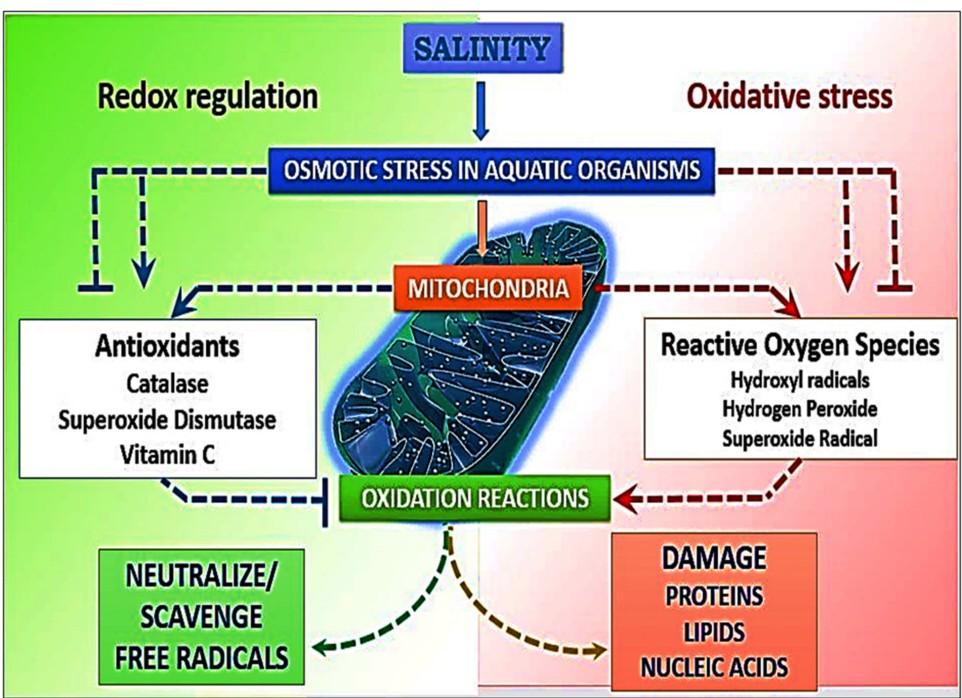

**Figure 5.** An overall outline of the oxidative stress responses and redox signaling in animals under salinity stress: Environmental osmotic stress can induce the generation of reactive oxygen species, but at the same time it also induces redox signaling. If the latter system is deactivated, it can lead to the alleviated expression of redox-regulating enzymes, resulting in elevated oxidative stress. Mitochondria play a pivotal role in this process.

Salinity can induce OS in organisms and, in contrast, the expression of antioxidant enzymes or responses in animals can contribute to augmented levels of amino acids [48,96,97]. Therefore, studying the intricate relationships between salinity and redox responses to balance or collapse the redox regulatory pathway is of paramount importance in aquatic systems. Since these phenomena are species-specific, a comparative literature review on the topic of OS responses to salinity in different phyla could fill the gaps in our understanding

of this issue. Therefore, for the first time, we tried to consider all available information on the effects of salinity on the OS responses in organisms belonging to different phyla.

### 3.2. Stress Incurred in Animals under Salinity Stress

The osmotic discrepancy at the cellular level that is caused by the altered salinity can accelerate the physiological functions. For normal functioning under salinity stress—i.e., by maintaining the osmotic balance and the ionic gradient—the specialized systems (e.g., excretory organs) demand higher energy expenditure in animals [98]. In addition, maintenance of the ionic gradients along with the gaseous exchange is quite important and, under the circumstances of altered salinity, specific organs such as the gills, canals, or skin of aquatic animals play important roles in osmoregulation and the associated physiology [99]. Fluctuations in salinity can cause spikes in ROS accumulation and OS in animals. To protect against ROS-induced stress under higher salinity levels, animals develop redox regulation signaling systems that are species-specific (Figure 5).

The signaling systems in animals (e.g., nuclear factor erythroid 2-related factor) work in coordination with the redox-regulatory enzymes [100]. Investigations of the OS, ROS, redox, and associated physiology under salinity fluctuations are greatly associated with ecological studies, which have gained remarkable attention in the last few decades [101] (Figures 2 and 3). The levels of ROS generation, enzyme activities, and the activity of small antioxidant molecules for redox regulation vary widely in different tissues and, furthermore, show variance in different species [102] (Tables 1–3). The altered physiology induced due to salinity shifts is correlated with the cellular responses and the different responses at the organismal level, which involve various physiological strategies. These include survival and reproduction, along with the moderated levels of transcription in different enzymes and corresponding variations in gene expression under various expected and unexpected saline environments.

Responses of Animals to Primary and Drastic Salinization

Under the primary salinization condition, in hyperosmotic and hypo-osmotic environments, the OS physiology is activated (Figure 4). In particular, low salinity is responsible for augmenting the expression of the redox-regulatory enzymes in *Dicentrarchus labrax* [103]. The number of red blood cells and white blood cells increases under extreme salinity conditions, and this can also cause deformities in erythrocytes with higher hemoglobin levels in European seabass (*Dicentrarchus labrax*) [104]. The altered salinity stress can also lead to differentially regulated genes in the yellowfin sea bream (*Acanthopagrus latus*) [47]. Therefore, increases in environmental salinity can be stressful to many organisms.

Salinity can greatly influence water-intake capacity to compensate for water loss under hypersalinity-induced osmotic imbalance in two species of *Culex* mosquitoes [105]. In contrast to the above outcomes, elevated levels of antioxidant defense have been reported in *Ruditapes decussates* and *Ruditapes philippinarum* in response to lipid peroxidation induced by hypersalinity [43]. Thus, periodic changes or predicted alternations in salinity can lead to a controlled response in the organism, whereas the scenario may be different in response to unpredictable or drastic changes in salinity.

Specific strategies are often adapted by the inhabiting organisms to combat salinity stress under altered climatic/seasonal changes. For example, the seasonal variation in salinity in the Chilika Lagoon of Odisha, India shows 0–5 ppt in the rainy season, ~17 ppt in winter, and ~35 ppt in summer [6] (Tables 1–3). Various strategies—such as raising the level of antioxidants, preparation for OS, lowering metabolism or metabolic depression, low respiration rates, low food uptake, and electron leakage in the ETC (coupled with low ROS production)—are adopted to reduce the chance of cells experiencing OS. However, when the change in salinity is sudden and/or drastic, it pushes the inhabiting organisms towards oxidative burst. Therefore, such ecophysiological conditions need to be addressed comparatively as a function of phyla, because the mechanism of osmosensing in mitochondria acts as the primary hub for the generation of ROS (Figure 6).

**Table 3.** Redox responses to counter OS in animals under salinity stress.

| Response | Species | Osmoregulation Hypo/Hyper/Iso | Salinity | Respiration Rate | ROS Responses | Redox Enzyme Level |
|---|---|---|---|---|---|---|
| Extracellular responses | *M. ligano* | Hyper/iso | Low salinity | ↑ | ↑ ($O_2^-$) $H_2O_2$ other ROS ↓ | ↑ |
| | *C. aestuarii* | Hyper/iso | High salinity | ↑ | ↑ | SOD ↑ |
| | *N. granulata* | Hyper/hypo | High salinity | ↓ | – | ↑ |
| | *S. serrata* | Hyper/hypo | Salinity | | | SOD ↑ CAT ↑ |
| | *B. koreanus* | ND | High salinity | – | ↑ | GST activity ↑ |
| | *A. naccarii* | ND | High salinity | – | ↑ | SOD ↑ CAT ↑ GPx ↑ |
| Intracellular responses | *A. microstoma* | ND | Low salinity | – | ↑ | CAT ↑ SOD insignificant |
| | *A. tamarensis* | ND | High salinity | – | ↑ | SOD ↑ CAT insignificant |
| | *C. gigas* | Osmoconformer | Wide range | No effect | NA | SOD not affected CAT ↑ at high salinity |
| | *D. labrax* | ND | Low salinity | – | ↑ | CAT↑ |
| | *L. vannamei* | ND | Both high and low salinity | – | ↑ | SOD ↑ CAT ↑ |
| | *A. schlegeli* | ND | Low salinity | – | – | SOD ↑ CAT ↑ |
| | *S. broughtonii* | ND | Low salinity | – | – | SOD ↑ CAT ↑ |
| | *S. plana* | Osmoconformer | Low salinity | – | TBARS ↑ | GPx ↑ GST↑ CAT ↑ |
| | *D. neopolitana* | Osmoconformer | Low salinity | – | – | CAT ↑ |
| | *V. decussate* | Osmoconformer | Wide range of salinity | – | LPO high | GSH ↑ SOD high at low salinity for all |
| | *V. corrugata* | ND | Change in salinity | – | LPO high | GSH ↑ CAT high at low salinity for all |
| | *V. philippinarum* | ND | Change in salinity | – | LPO high | GSH high |
| | *H. discus discus* | Osmoconformer | Lower salinity | – | – | ↑ |
| | *C. angulata* | Osmoconformer | Lower salinity | – | – | No change |
| | *R. philippinarum* | Osmoconformer | Lower salinity | – | – | SOD ↑ |
| | *C. gigas* | Osmoconformer | Lower salinity | – | – | No change |
| | *R. decussates* | Osmoconformer | Higher salinity | – | High LPO at extreme salinity | SOD ↑ CAT ↓ GST ↓ |
| | *R. philippinarum* | ND | Changing salinity | – | | SOD ↑ CAT ↑ GST no change |

The classification was performed based on respiration rates, levels of active oxygen molecules, and the activity or expression of redox-regulatory enzymes. The symbols ↑ and ↓ are used to denote elevated and reduced levels of the parameters, respectively. SOD: superoxide dismutase; CAT: catalase; GSH: the reduced glutathione; ROS: reactive oxygen species; LPO: lipid peroxide; GPx: glutathione peroxidase; GST: glutathione-S-transferase; ND: not detected; NA: not available. Modified based on the work of Bal et al. [7].

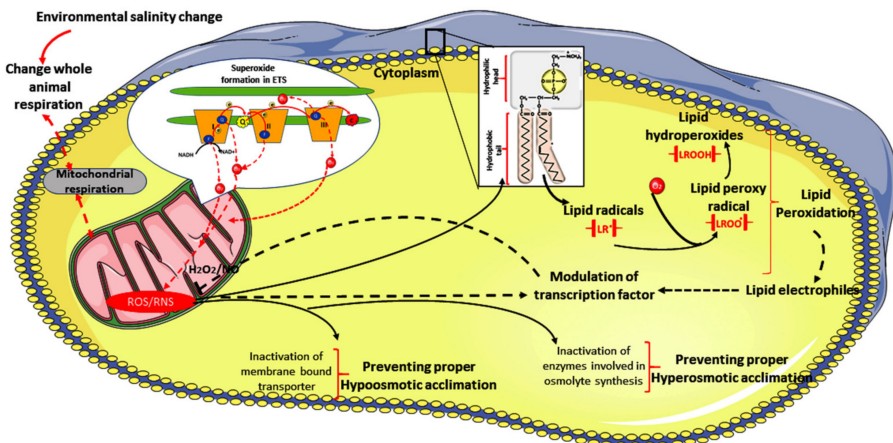

**Figure 6.** Responses of mitochondria to changing salinity: When mitochondria experience changes in the osmotic pressure in the cell due to environmental ionic strength, it may lead to electron transport chain dysfunction, thereby resulting in more electron accumulation in the intermembrane space. Thus, superoxide radicals are generated and are converted into $H_2O_2$ by the enzyme Mn-superoxide dismutase. The hydrophobic or hydrophilic nature of the mitochondrial membrane can allow or restrict inflow of lipids that leads to their oxidation. This establishes the first steps to generate oxidative stress, because lipid radicals are produced in the chain following their further oxidation.

### 3.3. Inclusive Responses of Organisms with Respect to Alterations in Salinity

Salinity in general modulates various biochemical and molecular events in helminthes. Marine nematodes are also not an exception to this rule. Recent work on the emerging marine nematode model *Litoditis marina* indicates the downregulation of multiple receptors responsible for neurotransmission and genes that regulate ion transport. Under hypersaline stress (30 ppt), the above receptors are downregulated. In contrast, the expression of genes that regulate unsaturated fatty acid biosynthesis, neuron-related tubulins, and intraflagellar transport is alleviated in the nematodes [106]. Therefore, redox status modulation and changes in OS physiology are apparent in nematodes under changing salinity.

Aquatic organisms, including helminthes, adjust themselves in terms of specific redox-regulatory strategies by maintaining different levels of redox-regulatory molecules to counteract the OS induced by salinity (Figures 3–5; Tables 1–3). Lower organisms, such as ectotherms (e.g., *Scylla serrata*), experience OS under altered salinity. In particular, high salinity induces OS in *S. serrata* due to insufficient redox capacity. However, the response can be tissue- or organ-specific [6] (Tables 2 and 3). Accordingly, the demand and supply of energy accompanied by redox regulation responses also vary among the animals [19–21].

It should be noted that salinity alone, or with multiple environmental stressors, can influence molecular mechanisms at the metabolomic, transcriptomic, proteomic, and genomic levels [107]. Therefore, salinity can potentially influence growth factors and reproduction at the biochemical and molecular levels, which may subsequently affect phylum-specific OS directly or indirectly in organisms and in a phylum-specific manner [19,104,108–113]. This may be due to switching of the signaling pathways that ultimately influence the OS physiology in aquatic animals under salinity stress (Tables 2 and 3).

#### 3.3.1. Rotifers

Rotifers are commercially cultivated to be served as food for the larvae of fish. Being microscopic in nature, rotifers generally reside in freshwater. Conversely, they are rarely found in marine water bodies. However, rotifer populations and OS mechanisms are adversely affected by increasing salinity, as is evident from a study conducted on the marine rotifer *Brachionus plicatilis*. The increase in levels of salinity up to 35 psu initiated a surge in the ROS levels and activated the activity of antioxidant enzymes such as SOD,

CAT, and GST [114]. The effect of salinity on the rotifers and the comparative signaling pathways in marine and freshwater environments is an unexplored avenue for researchers.

### 3.3.2. Helminthes

Hypersaline conditions lead to dehydration in worms, as water efflux facilitates the uptake of organic osmolytes in the environment [115]. However, under experimental conditions, when organisms are exposed to artificial seawater, the absence of osmolytes in their cells is noticed. This urges organisms to synthesize free amino acids to recover from osmotic stress during the initial state of osmoregulation, as observed in the case of the copepod *Tigriopus californicus* [116] and the shrimp *Penaeus aztecus* [117]. However, similar studies on helminthes are scarce.

In a study on *Macrostomum lignano*, it was observed that an increase in their respiration rate occurred under hypersaline stress. This was coupled with the formation of new mitochondria to meet the energy demand for the required adjustment (Figure 6). On the other hand, the study also observed an increase in the production of superoxide radicals under hypersaline conditions in *Macrostomum lignano*. The formation of ROS can have adverse effects on phospholipids and other components of cell membranes by causing lipid peroxidation and forming hydroperoxides in helminthes.

In contrast, a decline in respiration rates as well as in the production and consumption of ATP was observed under hypo-osmotic stress in *Procerodes littoralis* [118]. This leads to a hypoxic state in the organism, which eventually results in metabolic shutdown due to the oxidative burst [42]. Thus, the strategies adopted by organisms to increase their redox-regulatory status, as explained in "preparation of OS", seem to work in an effective manner in the above organisms.

### 3.3.3. Mollusks

Pulmonates are found to survive in both freshwater and seawater because they perform osmotic regulation, e.g., hypo-osmosis in freshwater and hyperosmosis in seawater [119]. Bivalves are able to tolerate wide fluctuations in salinity. The osmoregulation under extreme salinities is modulated by an increase in the calcium and sodium ion contents and shell valve closure. The closure of the shell valve leads to hypoxia and starvation under hypo-osmotic stress. Gradually, this leads to disturbances in the osmotic balance, which further allow the inflow of water into the cell, causing cellular rupture. Therefore, the environmental and bodily osmotic pressure is well mediated in mollusks (Figure 7).

In general, salinity modulates the expression of a vast number of enzymes and their respective genes in mollusks. Modulation of FoxO signaling, expression of tight junction proteins, and changes in immunomodulator titers under hyposalinity in Hong Kong oyster (*Crassostrea hongkongensis*) are a few examples. More specifically, hypersalinity-induced amino acid metabolism, along with signaling pathways such as AMPK and PI3K, is also recorded in the abovementioned oysters [120]. Similarly, both high (35 ppt) and low (10 ppt) salinity can induce increases in mRNA and amino acid levels in the hard clam (*Meretrix lusoria*), which possibly explains the upregulation of different proteins and genes under salinity stress in mollusks. The high energy dissipation cost of the salinity stress in mollusks is usually maintained, as observed in the Pacific abalone *Haliotis discus hannai*. The observed positive and strong correlation between cAMP activity, mRNA levels, and $Na^+/Ca^+$ ATPase in mollusks confirms this fact [121].

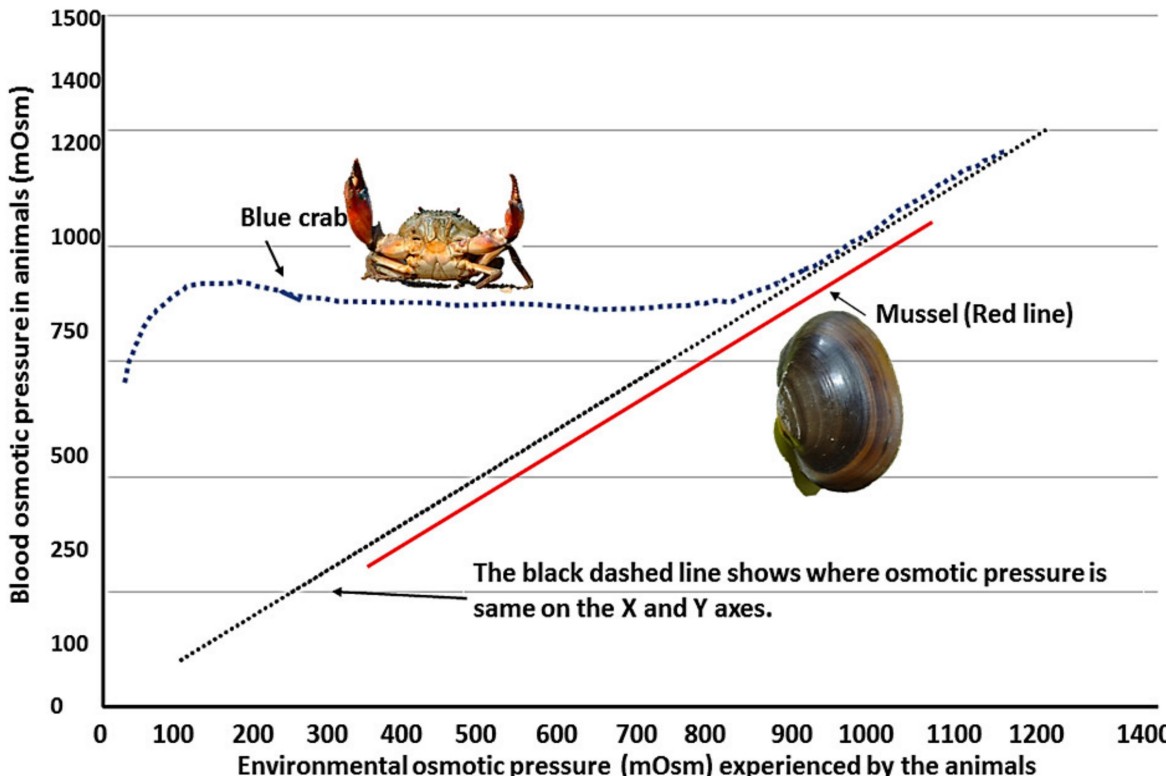

**Figure 7.** Effects of salinity on maintaining environmental and bodily osmotic pressure: The bodily osmotic pressure rises sharply under hypersaline states in osmoconformers such as mussels, while the reverse is observed in osmoregulators such as crabs, for example, which can regulate their bodly osmotic pressure up to 1000 mOsm under environmental osmotic conditions. However, higher environmental osmotic pressure under hypersalinity misbalances the osmoregulation process in many aquatic animals, irrespective of their osmoconformer status or regulatory capacity. The photo of the species was collected from Google Images with a creative common attribution license.

Changes in salinity and $CO_2$ levels are potential stressors that can induce OS in bivalves such as *Venerupis philippinarum* [122], *Ruditapes philippinarum* [123], *Pila globosa* [124–126], and *Crassostrea rhizophorae* [127]. The activity of the ETC is increased, followed by an increase in the lipid peroxidation level, thereby accelerating the metabolism under low salinity stress. To mask the toxic effects of lipid peroxidation, redox-regulatory enzymes such as SOD, CAT, and GST are activated. Higher salinity directly inhibits the SOD and CAT activity levels; conversely, the CAT activity is found to increase under lower salinity [128]. Therefore, the majority of the studied mollusks experience OS under high salinity stress. Recent studies have also provided contrasting results indicating that low salinity may also upregulate the cellular ROS levels of the shallow water clam *Scapharca subcrenata*, ultimately elevating the activities of Mn, SOD, and GPx in [129].

Salinity and salicylic acid (the constituting substance of pharmaceutical and personal care products) in combination cause minimal cellular damage. However, the metabolism of mollusks is enhanced in the presence of salicylic acid under optimal salinity conditions, but it declines with the increase in salinity [130]. On the other hand, changes in salinity can also be tolerated by certain mollusks, since no elevation in the oxidation of lipids is observed under higher-salinity conditions. For example, no significant changes in the lipid peroxidation levels are observed under different salinity regimes, whereas in some cases—as observed in *Ruditapes decussatus*, *R. philippinarum*, *Venerupis corrugata*, and *Crassostrea rhizophorae*—low salinity induces higher levels of lipid peroxides and antioxidant enzymes such as SOD, CAT, and GST [44,122,127].

Salinity-induced hypoxia reduces oxidative phosphorylation by restricting the production of ATP (due to less oxygen concentration in the cells) and activates alternative pathways for generating ATP that use less energy, as observed in protein catabolism in *Argopecten purpuratus* [131] and in *Ruditapes philippinarum* [132], for example. Therefore, the alternative oxidase system is one of the strategies maintained in animals to bypass complex III of the electron transport chain and to reduce the surplus production of active oxygen molecules [29] (Figure 4).

### 3.3.4. Annelids

The marine taxon Oligochaeta can adopt both hypo-osmotic and hyperosmotic regulation [119]. Studies on the salinity-induced changes in annelids are very scarce. A study was conducted on the exposure of *Neapolitan diopatra* to salinity. Under the conditions of hypersalinity (i.e., at 42 g $L^{-1}$), the lipid peroxidation levels were recorded to be the highest, whereas under hyposaline conditions (21 g $L^{-1}$) the lipid peroxidation level was comparable to the lowest. This was consistent with the elevated levels of activity of the antioxidant enzyme SOD and the biotransformation of the enzyme GST, and the reverse was recorded in the case of exposure to salinity of 28 g $L^{-1}$. The activity of CAT did not show any significant changes at varying salinity levels [37]. Therefore, an extensive study is essential to evaluate salinity-induced OS responses in annelids.

### 3.3.5. Arthropods

Salinity and $O_2$ levels in the environment are usually inversely proportional to one another. This may be an external cause for why crustaceans are found to suffer from hypoxia under conditions of higher environmental salinity. As a result, they shift towards hyperoxia when exposed to low salinity and, hence, consume more oxygen. The osmotic stress conditions arising due to salinity shifts lead to the accumulation of different osmolytes in the tissues involved in the process of osmoregulation in shrimps. $O_2$ consumption and osmotic imbalance regulate OS in animals, and crustaceans are not an exception to this rule; however, they can shift their niche to find a suitable salinity regime (Figure 7).

Delgado-Gaytán et al. [46] reported that the activity of the osmolyte betaine aldehyde dehydrogenase was increased in the gills and hepatopancreas of white shrimp at higher salinity [46], thereby leading to an increase in betaine glycine content in these tissues. Remarkably, this also affected the ammonia content, which showed a significant decreasing trend at higher salinity levels. The reported increase in the rate of $O_2$ consumption at lower salinity levels should generate a higher energy budget, which further facilitates the passage of ions into the hemolymph from the habitat. In contrast, at higher salinity levels, the oxygen consumption is reduced, along with the lowering of the oxidative metabolism of the organism. The above phenomenon was confirmed by a study on South American crabs (originating from the sea but residing in freshwater). The species retained its ancestral osmoregulatory mechanism when exposed to higher salinities [133]. The adaptation to combat higher salinity stress was accompanied by a rise in hemolymph osmolality and a decline in V-H$^+$-ATPase activity. Thus, energy demand and supply are highly correlated with the osmotic regulation and OS responses in crabs.

Experiments on *S. serrata*—a euryhaline crab—confirmed the reduced levels of lipid peroxidation in various tissues at lower salinity, accompanied by elevated levels of antioxidant enzymes such as SOD, CAT, and GPx in the gills and only SOD in the abdominal muscle tissues [6]. Thus, the mechanism in this crab suggests that the higher consumption of oxygen in the above tissues at lower salinity could lead to an increase in the production of ROS such as lipid peroxides, $H_2O_2$, and superoxide radicals, causing an increase in antioxidant levels. During the hypoxia state, when the species are subjected to higher salinity, it reduces the transport of electrons in the ETC [6,20]. Thus, euryhaline crabs may experience OS under both hypoxic and hyperoxic conditions that align with hyper- and hyposaline regimes in the environment.

This elicits the formation of ROS from electron carriers such as complex III ubiquinone. Among the tissues of crabs, the abdominal muscles were found to have lower metabolism with respect to the hepatopancreas and gills, leading to the conclusion of lower levels of superoxide radical production [6,134] (Tables 2 and 3). Several other studies carried out on crustaceans such as *Macrobrachium rosenbergii* [135], *M. malcolmsonii* [136], *Charybdis japonica* [137], *Neohelice granulate* [138,139], *Oronectus limosus* [140], *Carcinus maenas* [141], and *Litopenaeus vannamei* [47] have reported similar increases in ROS production under salinity stress. Confirmation of the modulation of OS in arthropods under altered salinity can be obtained from the above examples.

### 3.3.6. Echinoderms

Within the span of evolutionary development, the echinoderms and ascidia have not been able to overcome the physiological adjustments in freshwater. The echinoderms are less adapted for adjusting to osmotic changes under salinity fluctuations. This is because they are devoid of the proper organs that can regulate osmotic balances. Thus, the coelomic fluid that is isosmotic to the environment is affected by salinity fluctuations [142,143]. Therefore, these organisms may become highly susceptible to OS under salinity stress.

The oxygen consumption rate is considered to be one of the most important parameters for energy production. In echinoderms such as *Apostichopus japonicus* (energy budget), *Ophiophragmus filograneus* (respiration), and *Holothuria grisea* (ionic gradients and hydration issues), the rate of oxygen consumption is altered under hypersaline conditions [144–146]. In *Holothuria leucospilota*, a decline in energy production is noticed under low saline exposure. Higher $O_2$ consumption is observed in many organisms under low salinity. Higher $O_2$ consumption is accompanied by high oxidative phosphorylation. The opposite trend is observed in *H. leucospilota*, setting up a new direction for further investigations.

The decline in the $O_2$ consumption rate consequently leads to a reduction in metabolic activity. This indicates that adverse effects on specific tissues can potentially generate higher levels of ROS under low-salinity conditions [147]. Thus, metabolic depression may further induce elevated levels of antioxidant defenses that may help in the organisms' recovery from oxidative bursts [48]. Echinoderms can be a good model to elicit the above fact.

### 3.3.7. Fishes

Fish can thrive in both freshwater and seawater. They adapt to both hypo-osmotic and hyperosmotic regulation in freshwater and seawater, respectively [119]. Salinity changes in inland freshwater and coastal wetlands extensively affect the wild habitats of freshwater species such as *Catla catla*, *Labeo rohita*, *Cirrhinus mrigala*, and *Tenualosa ilisha* in various ways [148]. For example, species specific to positive or negative growth rates may be observed under low-salinity conditions [149–151]. Thus, salinity fluctuations are one of the critical reasons behind the growth performance of fishes.

Specific fish species also show unique patterns in responsive OS events under salinity stress. Significant upregulation of $Na^+/K^+$ ATPase$\alpha$1, $Na^+/K^+$ ATPase$\alpha$2, and $Na^+/K^+$ ATPase$\alpha$3 was observed in the white-rimmed stingray *Himantura* sp. when exposed to saline stress. This fish also showed higher $Na^+/K^+$ ATPase$\alpha$3 mRNA levels and proteins of $Na^+/K^+$ ATPase in general under salinity stress [152]. Energy homeostasis can be maintained by a heavy rate of catabolism without changing the levels of glycogen in the livers of fish under salinity stress, as observed in the milkfish *Chanos chanos*. However, the $Na^+/K^+$ ATPase in the larvae of *Takifugu obscurus* increased significantly along with the levels of malondialdehyde and SOD activity with an increase in salinity [153]. This indicates the tolerance adaptation strategy in the fish under salinity stress [154].

Modulation of antioxidant enzymes and OS events under changing salinity is clear in fishes. Low salinity (5 ppt) stress for up to 144 h in the yellow-striped American perch (*Perca flavescens*) alleviates the expression of CuZn SOD, GPx, and GST, along with Hsp70 [155]. In addition to stenohaline fish, the physiological disturbances are also espe-

cially evident in euryhaline fish such as silver sea bream when exposed to salinity levels of 0, 6, 12, 33, 50, and 70 ppt [156]. The most important molecules that respond and play a vital role with respect to changing osmolality are aquaporin in general and aquaporin Ia in the gills, kidneys, intestines, and mantle tissues, which have now been correlated with the expression of regulatory redox and energy homeostasis genes [7].

Hypo-osmotic shock is associated with oxidative injury due to low antioxidant status in fish. Under lower salinity, the higher rate of oxygen consumption leads to the formation of surplus ROS and free radicals, and reduced antioxidant defense activity leads to an imbalance in the redox state [157,158]. In a detailed study on *Pseudosciaena crocea*, it was observed that salinity stress induces elevated expression of an antioxidant signaling marker in the nucleus (nuclear factor erythroid 2-related factor 2), which suggests that antioxidant genes were upregulated under low-salinity conditions [159,160]. The augmented expression of this nuclear marker in the nucleus is usually beneficial in cells, as it signals for antioxidant enzymes. Sometimes it includes subsequent events of apoptosis and tumors due to free radical damage. To maintain the optimal level of erythroid 2-related factor 2 in the nucleus, as a protective role, KEAP1 expression is activated as a control measure over its excess production. In addition, under salinity stress, the nuclear factor erythroid 2-related factor 2/Kelch-like ECH-associated protein 1 pathway plays a pivotal role in upregulating the redox regulation via the expression of redox-regulatory enzymes [100,161].

Under hypersaline conditions, OS was induced, as evidenced by elevated levels of lipid peroxidation and an inability of the antioxidant defense system to produce enough antioxidant molecules to eliminate ROS [162]. Salinity-induced stress leads to demand and supply of energy that allows animals to adapt to an optimal metabolic rate by activating the enzymes that take part in glycolysis and oxidative phosphorylation [163]. Different studies have established that salinity mainly affects the mitochondria of gills in fishes due to stress, which further generates ROS. In some species, such as *Acipenser naccarii*, the activated osmoregulatory process led to an increase in lipid peroxidation levels in the blood, which was not able to revert to its usual levels even after normal conditions were restored [164]. Although salinity has detrimental effects on OS physiology, the idea of dietary supplementation for fishes in adverse saline conditions could be an effective intervention to enhance their physiological mechanisms. Dietary supplementation of *Aspergillus oryzae* in the food of the Nile tilapia significantly enhanced the antioxidative enzyme pathway by increasing the activity of enzymes such as SOD, CAT, and GPx under increasing salinity stress. The dietary supplement was also considered to be effective in enhancing the non-specific immune responses and blood protein profiles [165]. Thus, a dietary substitution could be an effective implementation to overcome the adverse stress conditions caused by salinity in fishes.

It has also been observed that, along with multiple stressors—especially heavy metals—salinity can modulate the OS in various fish models. Studies carried out on the euryhaline killifish *Fundulus heteroclitus* as a function of zinc, copper, and nickel by Loro et al. [166], Blewett et al. [167], and Ransberry et al. [168], as well as by Glover et al. [169] on the galaxiid fish *Galaxias maculatus*, have observed that salinity extensively modulated the OS physiology. Thus, it can be concluded that fish, being higher-taxon organisms, can activate various signaling systems such as the nuclear factor erythroid 2-related factor 2/Kelch-like ECH-associated protein 1 system to control OS-mediated salinity, but sometimes the collapse of the system may lead to OS in the fish. Therefore, studying the signaling system seems important in the present context.

From the above discussion, it is clear that the OS responses in aquatic animals under changing salinity are species-specific. All of the organisms follow their own specific strategies for osmotic balance in relation to OS responses and the energy demand and supply chain of their cells (Figure 8). Therefore, when any specific aquatic models are used for research or aquaculture purposes, water salinity levels must be considered as one of the important physicochemical factors.

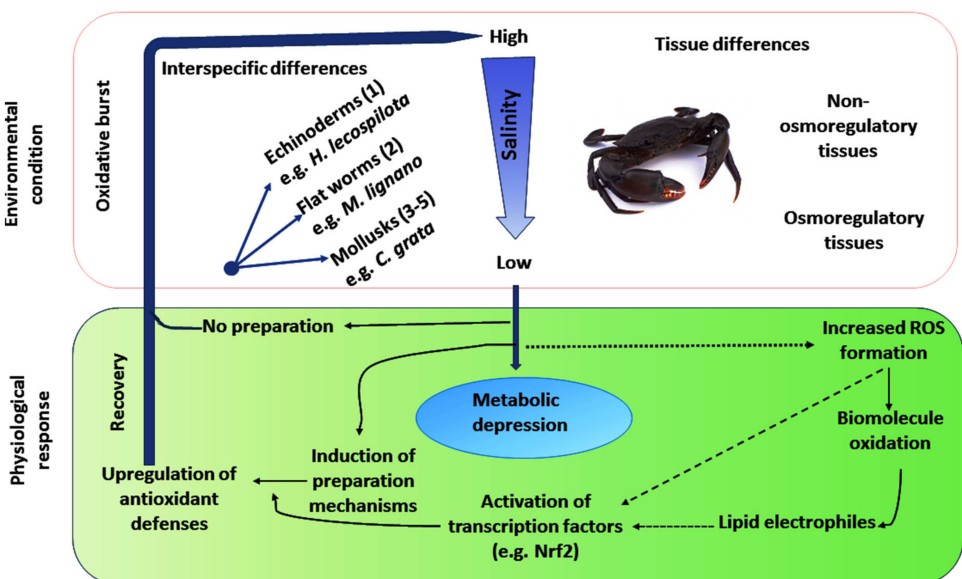

**Figure 8.** Redox-regulatory responses in different phyla under salinity stress: Different organisms have comparatively different adaptive mechanisms to control bodily osmotic pressure. The final osmotic tissue pressure under low or high salinity invariably leads to high reactive oxygen species in all lower invertebrates, including mollusks, worms, and echinoderms. NrF2 signaling to sense the high ROS-induced redox regulation via antioxidant enzyme expression may be able to control the reactive oxygen species and oxidative stress, and other animals may undergo low metabolic adaptations such as metabolic depression that can help the animals to produce low amounts of reactive oxygen species and achieve high rates of survival under salinity stress.

### 3.4. Signaling Pathways under Salinity Stress

High salinity increases water loss from the body and also facilitates the impairment of turgor pressure. As a result, osmolytes are taken in, as high salinity may be involved with the degradation of proteins and regulating cell volume through the regulation of associated genes and apoptosis [170–173]. For effective regulation of cell osmoregulation in the inhabiting organism, including fish under changing salinity conditions, signaling molecules such as phospholipase A2, calcium-sensing receptors, and cytokine receptors are activated. This aspect requires deep studies in aquatic models [174].

In this context, the role of respective hormones and receptors is well documented under changing salinity stress [175,176]. The protein phosphorylation events such as stimulated protein kinase signaling pathways that are activated by mitogens in different fishes were greatly influenced by the changes in salinity levels—for example, in tilapia [86], killifish [177], and turbot [87]. Similarly, several other factors—such as the activation of light chain kinase of myosin, focal adhesion kinase, and osmotic stress transcription factor—have been reported in different species i.e., the Japanese eel [91], killifish [87], and osmotic transcription factor 1 in *Mozambique Tilapia* [88], as systematically documented in this review (Table 3). This aptly indicates the activation of several signaling pathways that have evolved under changing salinity levels in aquatic models in general and in fish in particular.

Negative effects of salinity on signaling pathways have also been observed in some animals. The euryhaline bottom-dwelling goby *Neogobius melanostomus* can turn down its osmoregulatory activity at or above 25 ppt salinity. Thus, higher levels of blood plasma osmolality compete against the survival rate [178]. Higher energy expenses are borne by the animals due to activated levels of $Na^+/K^+$-ATPase activity when the animals shift from low to high salinity [179], and investigations in *Fundulus heteroclitus* [180] and *Oreochromis mossambicus* [181] validated the above fact experimentally. However, organisms try to use the signaling pathways judiciously to avoid harsh environmental conditions

under changing salinity stress. This is because the adverse impact of salinity has already been established in invertebrates [20]. Simple seasonal changes in salinity may also impart such stress and activation of the redox sensing system to upregulate the associated enzyme responses (Table 2).

Aquatic organisms are reasonably sensitive to variations in salinity, which may be seasonal or drastic and unpredictable. Organisms from the lower phyla with comparative less motility capacity, such as *Perna viridis*, can experience more stress under salinity fluctuations [182]. Sometimes such organisms express lower redox-regulatory capacity in the form of lower activity of SOD, CAT, and GPx or high but insufficient expression of redox-regulatory enzymes. This may lead to the accumulation of cellular thiobarbituric-acid-reactive substances, as observed in the pearl cichlid (*Geophagus brasiliensis*), the three-spined stickleback (*Gasterosteus aculeatus*), and the Senegalese sole (*Solea senegalensis*). The regulation of OS physiology in many organisms—such as *G. brasiliensis, G. aculeatus, S. senegalensis*, and *B. barbus* is influenced by several stress factors, such as temperature and salinity, in combination or alone [183].

In the above organisms, the mitochondrial functions and energetics are changed accordingly, and this may occur under both low and high salinity stress, as both produce high ROS in different aquatic animals (Figures 3–6; Table 1). This is also linked to elevated metabolic rates in animals, which are usually coupled with elevated levels of ROS generation in cells, as observed in *M. lignano* and *Saccharomyces cerevisiae* [32,48]. Therefore, the involvement of other environmental parameters such as $O_2$ levels and temperature, which are directly linked to salinity, may contribute to the altered OS physiology in different phyla. However, lower organisms have developed fewer strategies to cope with such changes. Salinity, when combined with anthropogenic factors, causes more species-specific impacts on the inhabitants (Table 4), and knowing the sources of the anthropogenic factors and their impacts on the salinity of the habitats and the physiology of their inhabitants will be helpful for the management of aquaculture resources.

**Table 4.** Effects of anthropogenic factors on changes in salinity in coastal areas.

| Anthropogenic or Climatic Factor | Area of Investigation | Impact on Salinity | Reasons for Salinity Alteration | Reference |
|---|---|---|---|---|
| Temperature and rainfall | Khorezm, Uzbekistan | Increased salinity | Decreased groundwater resources | [184] |
| Glacial melting | Indian Sundarbans | Decreased salinity in the Hooghly estuary | Freshwater input from Himalayan glaciers | [185] |
| Regional warming | Central Gangetic Delta, India | Increasing salinity | High rate of evaporation | [186] |
| Global warming | NA | Enhanced salinization | Nutrient loading, evaporative up-concentration of nutrients in reduced water volumes | [187] |
| Wind direction | Coast of Tarragona (Spain), Uruguayan coast | Increases atmospheric salinity | Entrainment of marine aerosols from the sea | [187–189] |
| Precipitation | Wuliangsuhai Lake, China | High salinity | Concentrations of salts increase with low precipitation | [190] |

**Table 4.** *Cont.*

| Anthropogenic or Climatic Factor | Area of Investigation | Impact on Salinity | Reasons for Salinity Alteration | Reference |
|---|---|---|---|---|
| Moisture | South Africa, Mexico, Germany, and Hungary | Increases salinity | Rock weathering releases mineral salts | [191–193] |
| Thermohaline circulation | Across the Atlantic to 47° N | Salinity increases of 0.38 psu | Cooling results in a density increase | [194] |
| Horizontal distribution | Equator (between 20–30 degrees north) | 34 ppt salinity | High humidity and less precipitation | [188,195] |
| Vertical distribution | 300–500 m depth | 33–37 ppt salinity | NA | [196] |
| Gaseous pollutants | Madrid | Increased salinity | Crystallization and hydration pressure; salt weathering | [197] |
| Sulfate/sulfuric acid | Madrid | Increased salinity | Crystallization and hydration pressure; carbonate and silicate rock | [197–199] |
| Nitrate/nitric acid | NA | Increased salinity | Decreased nitrate reduction, organic waste, and deicing salts | [197] |
| $Ca^{2+}$, $Mg^{2+}$, $SO_4^{2-}$, $HCO_3^-$, $Na^+$, and $Cl^-$ | Southwest Bengal Delta of Bangladesh | High salinity in both ground and surface water (7.5 to 8 ppt) | Saltwater shrimp cultivation, embankments, and excessive irrigation | [200,201] |
| Biocrusts | Alagol Lake, Northern Iran | Increased salinity up to 50% | Lack of biocrust induces high rates of evaporation | [202] |
| Manmade/natural | Yangtze River, East China Sea | Groundwater salinization | Seawater intrusion | [202,203] |

## 4. Conclusions and Future Prospects

Salinity is one of the major decisive environmental factors that modulate the physiology of the inhabitants of marine environments. Salinity of the environment is also modulated by various anthropogenic and climatic factors. Finally, the effects of environmental salinity on different physiological strategies—including growth and reproduction at the proteomic and genomic levels—have been studied in aquatic models using several genomic, proteomic, transcriptomic, and metabolomic tools over the last 50 years. The main goal may be to increase aquatic productivity to achieve "Zero Hunger" by 2030. Therefore, management of stress biology in general (and OS biology in particular) under salinity fluctuations as a function of phylum needs to be reviewed, because this information is scarce. However, OS biology is related to the physiology of aquatic organisms associated with their growth, production, and reproduction. Specific strategies for specific organisms need to be adapted from their stress response management under fluctuations in salinity. Organisms from lower phyla may respond sharply to a narrow range of salinity fluctuations, as they have fewer molecular adaptive mechanisms, while higher organisms such as fish may show better responses to salinity stress (Figure 9). According to their OS, their physiology is modulated and, hence, can be considered as markers for comparative studies between organisms for their stress tolerance. Responses at the mitochondrial level and energy expenditure are also coupled with salinity stress. The physiological adaptation strategies of marine aquatic organisms—including both vertebrates and invertebrates—in response to particular levels of environmental salinity are evident from significant studies at specific tissue levels and organismal levels. Meanwhile, comparative studies on the responses of different phyla to salinity remain unexplored. Studies on OS and redox regulation by enzymes and non-enzymes in aquatic organisms have been conducted by various researchers as a function of biotic factors—especially salinity, which is significant for the survival of

the organisms belonging to various phyla—and these works have been validated by the molecular-level study of some factors that enhance the expression of antioxidant enzymes, such as the nuclear factor erythroid 2-related factor 2/Kelch-like ECH-associated protein 1 signaling system. This review systematically describes the OS physiology under changing salinity levels in different phyla for the first time and, thus, presents a clear picture of the different climatic and anthropogenic factors that sustain life underwater.

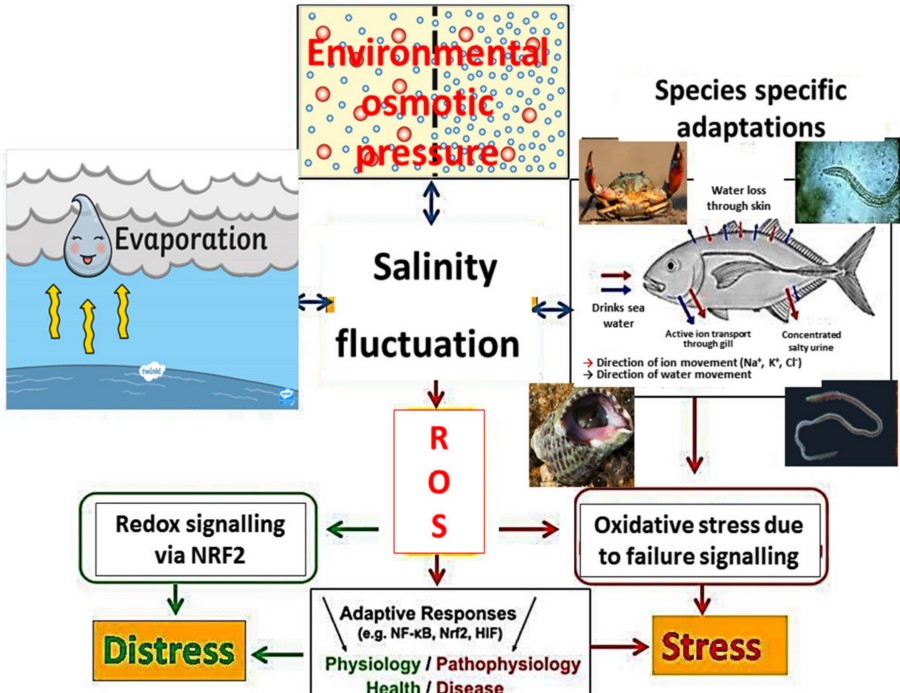

**Figure 9.** Redox regulation under salinity stress in animals: Low or high salinity can induce oxidative stress in aquatic animals, mainly due to the lower activity levels of the redox-regulatory enzymes. Hypersaline stress is associated with elevated oxidative stress in animals and explained through the high energy demand that needs speedy oxidative phosphorylation coupled with a high rate of the electron transport chain, which may lead to high electron leakage at complex I and III enzymes to produce more reactive oxygen species. Activation of the alternate oxidase system to leak fewer electrons via complex enzymes under hyperoxia (low salinity) is one of the strategies that may produce low amounts of reactive oxygen species under low salinity stress, as observed in the ectotherm *Scylla serrata* [6,19–21,29]. Metabolic adjustments in both osmoconformers and osmoregulators can be seen to protect against salinity stress, or the animal may simply migrate to an area with more favorable salinity (animals in inserts). Elevating the number of mitochondria is also one of the main strategies observed to meet the energy demand under salinity stress. Involvement of the nuclear factor erythroid 2-related factor 2)/Kelch-like ECH-associated protein 1 redox signaling system for redox sensing and upregulation of the redox-regulatory enzymes is now established. Balancing the above pathways enables some animals to survive and gives them the capacity to face challenges under a wide range of salinities (euryhaline), while some simply migrate to a safer salinity zone because such animals can tolerate very low fluctuations in salinity (stenohaline). Some open-access photos were collected from Google Images.

**Author Contributions:** Conceptualization, data curation, formal analysis, funding acquisition, investigation, methodology, project administration, resources, software, supervision, validation, visualization, writing—original draft, writing—review and editing, B.P.; conceptualization, writing—original draft, methodology, writing—review and editing, A.B.; writing—review and editing, revision, F.P., S.G.P., T.N.A. and K.D. All authors have read and agreed to the published version of the manuscript.

**Funding:** This work was generously supported by the funding to B.R.P. from the Science and Engineering Research Board, Department of Science and Technology, Govt. of India, New Delhi, India (No. ECR/2016/001984), and by the Department of Science and Technology, Government of Odisha (Grant letter number 1188/ST, Bhubaneswar, dated 01.03.2017, ST-(Bio)-02/2017). And funding Department of Science and Technology, Government of Odisha, India (1264/ST/BT-MISC-0034-2018) to AB to Abhipsa Bal under Biju Patnaik Research Fellowship to pursue Ph.D. course in Biotechnology is acknowledged.

**Institutional Review Board Statement:** Not applicable.

**Informed Consent Statement:** Not applicable.

**Data Availability Statement:** All data generated or analyzed during this study are included in this published article.

**Acknowledgments:** The authors duly acknowledge the use of the Central Instrumentation Facility of Odisha University of Agriculture and Technology, and especially the assistance of Sashikanta Dash in the analyses of samples.

**Conflicts of Interest:** The authors declare no conflict of interest.

## Abbreviations

PSU: practical salinity unit; ppt: parts per thousand; ROS: reactive oxygen species; OS: oxidative stress; SOD: superoxide dismutase; CAT: catalase; GPx: glutathione peroxidase; GSH: reduced glutathione.

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
