# Peer review of "Influence of Anthropogenic Activities on Redox Regulation and Oxidative Stress Responses in Different Phyla of Animals in Coastal Water via Changing in Salinity"

_water, doi:10.3390/w14244026_

Round 1
Reviewer 1 Report
Below there are just a few remarks related to the quality of the submitted manuscript. From the beginning it became evident that the proposed review article is full of flaws. Thorough revision with support of acknowledged experts is required should the authors want to resubmit a revised version:
ABSTRACT and INTRODUCTION
“…dissolved substances such as sulfate, sulfuric acid, nitrate, nitric acid, Ca2+, Mg2+, SO4 2−, HCO3−, Na+ and Cl− and biocrust…
R: Substances are not ions; sulfate, nitrate are anions, sulfuric acid, nitric acid are substances. Also, the authors write some chemical species with their full names and others with their chemical formulae; this should be harmonized.
L44-46- “Knowing the sources of the changing salinity especially in coastal belts is always useful for maintenance of the habitat and management of the affected species. It is because salinity influences the physiology of the organism to a greater extent”
R: The second sentence (It is….extent) does not stand by itself. Since it may pertain to the first sentence, they should be harmonized together.
L49- “millions of species distributed”
R: Which kind of species, chemical species, biological species or….?
L48-59- R: Revise completely. The whole paragraph is a mess, with different approaches to salinity mixed up, with the wrong definitions of quantities, units and their symbols, etc. Bibliographic references and contents of standard organizations and reference documents are missing and are a must.
L48- “About 90% of water bodies are saline”; L54-55-“Freshwater salinity remains ”.
R: If 90 % are saline, the other 10 % are not (???!!!). So does it make any sense to mention salinity of freshwater? Everything needs strong revision, of both contents and grammar.
…..
L154- al.,2022;
R: et al., 2022
L155.- “Now days”
R: Nowadays
….
L708- “This review opens up the way for new investigations that still remain unexplored.”
R- Which new investigations?
Author Response
Replies to Reviewers 1
Below there are just a few remarks related to the quality of the submitted manuscript. From the beginning it became evident that the proposed review article is full of flaws. Thorough revision with support of acknowledged experts is required should the authors want to resubmit a revised version:
Reply: We have now revised the article thoroughly and the conceptual and grammatical errors are now fixed.
ABSTRACT and INTRODUCTION
“…dissolved substances such as sulfate, sulfuric acid, nitrate, nitric acid, Ca2+, Mg2+, SO4 2−, HCO3−, Na+ and Cl− and biocrust…
R: Substances are not ions; sulfate, nitrate are anions, sulfuric acid, nitric acid are substances. Also, the authors write some chemical species with their full names and others with their chemical formulae; this should be harmonized.
Reply: We have harmonized the chemical names. And the line was changed to ….. dissolved substances and chemicals/ions such as biocrusts, sulfate, sulfuric acid, nitrate, nitric acid, Ca2+, Mg2+, SO42−, HCO3−, Na+ and Cl− are found to be…..
L44-46- “Knowing the sources of the changing salinity especially in coastal belts is always useful for maintenance of the habitat and management of the affected species. It is because salinity influences the physiology of the organism to a greater extent”
R: The second sentence (It is….extent) does not stand by itself. Since it may pertain to the first sentence, they should be harmonized together.
Reply: Now the lines are changed to “Exploring the source(s), mode(s), mechanism(s) and pattern of the changing salinity in water bodies, especially in coastal belts, is always useful for maintenance of the habitat and management of the inhabitants. It is because salinity of water largely influences the physiology of the inhabitants (Williams, 2001; Kefford et al., 2012; Sowa et al., 2019). “
L49- “millions of species distributed”
R: Which kind of species, chemical species, biological species or….?
L48-59- R: Revise completely. The whole paragraph is a mess, with different approaches to salinity mixed up, with the wrong definitions of quantities, units and their symbols, etc. Bibliographic references and contents of standard organizations and reference documents are missing and are a must.
L48- “About 90% of water bodies are saline”; L54-55-“Freshwater salinity remains ”.
R: If 90 % are saline, the other 10 % are not (???!!!). So does it make any sense to mention salinity of freshwater? Everything needs strong revision, of both contents and grammar.
Reply: We have modified the entire paragraph as “Salinity is measured as practical salinity unit (PSU), gL-1, parts per thousand (ppt) etc. Salinity is the saltiness of water, so it is depends on the concentration of salts in soluble form and thus is experimentally measured in terms of salt concentration. For example, it can be measured by quantifying the concentration Cl- (Robinson 1954; Haumann et al. 2016). The salinity of freshwater usually found to be between 0.5 to 1 ppt, while it ranges from 0.5 to 30 ppt in brackish water and its value is > 30 ppt in sea water. However, the salinity of some water bodies can be saturated up to 400 ppt (Fig. 1). Large saline- water bodies and fluctuation in their salinity ranges on the earth surface poses different impacts on the inhabitants staring form growth and reproduction to aquaculture production. It encourages researchers to work on the different aspects of animals and plants with respect to change in environmental salinity and its impacts on orgasms (Paital and Chainy 2010; Bal et al. 2021).
…..
L154- al.,2022;
R: et al., 2022
Reply: We have placed a space between et,, and 2022
L155.- “Now days”
R: Nowadays
Reply: We have corrected it as “Nowadays”
……
L708- “This review opens up the way for new investigations that still remain unexplored.”
R- Which new investigations?
Reply: We have deleted the line and modified its previous line as
This review systematically describes for the first time on the OS physiology under the changing salinity in different phyla and thus brings in a clear picture of the different climatic and anthropogenic factors that sustain life under water.
Besides the above comments, we have modified the entire manuscript and fixed all the issues in concept and grammar.
Reviewer 2 Report
This is an important review with a deep analysis (in the scopes of the available information) and very useful figures. I only recommend to check again the English grammar (I give an examples of incomplete sentences and absence of ‘comma’
Some specific comments are below with the abbreviation L (Line)
Title. Anthropogenic activities induced change in salinity in coastal water and their influence on redox regulation and oxidative stress in different phyla of animals
Unclear: which influence: Anthropogenic activities (in this case ): Anthropogenic activities influence on redox regulation and oxidative stress in different phyla of animals;
Or: If ‘change in salinity influence on redox regulation and oxidative stress in different phyla of animals ( why their – must be ‘its’
I recommend change to: Anthropogenic activities influence redox regulation and oxidative stress responses in different phyla of animals in coastal water via change in salinity
Abstract
L16 reject ‘substantially (you mentioned ‘decisive’ -it is enough)
L 23 (in)directly? Directly or indirectly
L 23 ‘influence the biomarkers of oxidative stress (OS) physiology (? – physiology is needless)
L 25 why ‘studied’ – past?
L 26 – correct the sentence (what scanty?)
LL 28-29 Correct grammar: ‘knowing the mechanism of response may be useful for the management of specific aquatic animal(s) or their habitat’.
Key words ‘; Aquatic invertebrate and vertebrates’ – better, to separate: ; Aquatic invertebrate and Aquatic vertebrates or use ‘aquatic animals’
Introduction
L 35 ‘is also’ – reject ‘also’
L 78 – OS, L 81 -ROS – decipher at the first mentioning
L 85 – not only for ATP production but also for the metabolite oxidation (microsomal oxidation reactions, all together – cellular respiration)
L 116 – use ‘OS’ instead of ‘oxidative stress’
L 118 – correct to Matozzo (Valerio Matozzo) here and in the reference list (#119)
2 Anthropogenic activities and climatic factors….. (influence?)
L 156 – change ‘Now days,’ to nowadays
LL 168,169 ‘where’ or ‘when’?
L 271 ‘Oxidative stress’ change to OS here and in other places after the first mentioning of ‘OS”
L 272 Why ‘Therefore,’ in this place?
L 275’ catabolism by kinases are’ - catabolism is – correct, please
L 277 – can you give an examples of this difference?
L 281 - '2003) and – put comma
L 287 – ‘blood osmotic pressure’ – blood or/and hemolimph?
L 322 -3.2.1. Response of animals to primary and drastic salinization - subdivision impossible (where 3.2.2?)
L 363, 346, 313 – tables, not Table 1-3; L 279 – why here you separate: (Table 1, Table 2).
L 693 – unclear: ‘According to their OS physiology is modulated’ (what is modulated?)
Tables 1-3 Please, provide the references;
Fig 1-2 Please, provide the references;
Author Response
Replies to Reviewers 2
This is an important review with a deep analysis (in the scopes of the available information) and very useful figures. I only recommend to check again the English grammar (I give an examples of incomplete sentences and absence of ‘comma’
Reply: We are thankful for the positive comments of the reviewers and beg apology for the grammatical mistakes. We have corrected the ms with a native English speaker.
Some specific comments are below with the abbreviation L (Line)
Title. Anthropogenic activities induced change in salinity in coastal water and their influence on redox regulation and oxidative stress in different phyla of animals
Unclear: which influence: Anthropogenic activities (in this case ): Anthropogenic activities influence on redox regulation and oxidative stress in different phyla of animals;
Or: If ‘change in salinity influence on redox regulation and oxidative stress in different phyla of animals ( why their – must be ‘its’
I recommend change to: Anthropogenic activities influence redox regulation and oxidative stress responses in different phyla of animals in coastal water via change in salinity
Reply: Yes, we agree with the reviewers comments. An after going through all of the suggestions, we have modified the title as “Influence of anthropogenic activities on redox regulation and oxidative stress responses in different phyla of animals in coastal water via change in salinity
Abstract
L16 reject ‘substantially (you mentioned ‘decisive’ -it is enough)
Reply: We have deleted the word.
L 23 (in)directly? Directly or indirectly
Reply: We have changed it to directly or indirectly
L 23 ‘influence the biomarkers of oxidative stress (OS) physiology (? – physiology is needless)
Reply: Yes, therefore we have delete d the term physiology
L 25 why ‘studied’ – past?
Reply: We have changed it to it is being studied
L 26 – correct the sentence (what scanty?)
Reply: We have changed the line as “Salinity modulated OS and redox regulation as a function of phylum is covered in this review.”
LL 28-29 Correct grammar: ‘knowing the mechanism of response may be useful for the management of specific aquatic animal(s) or their habitat’.
Reply: We have put this line in the suggested form at the end of the first para of introduction.
Key words ‘; Aquatic invertebrate and vertebrates’ – better, to separate: ; Aquatic invertebrate and Aquatic vertebrates or use ‘aquatic animals’
Reply: Thank you for the suggestion. We have changed it to aquatic animals’
Introduction
L 35 ‘is also’ – reject ‘also’
Reply: We have removed the word also.
L 78 – OS, L 81 -ROS – decipher at the first mentioning
Reply: We beg apology for this mistake and mentioned their full form at their first mention in the main text.
L 85 – not only for ATP production but also for the metabolite oxidation (microsomal oxidation reactions, all together – cellular respiration)
Reply: We have changed the line as per the suggestion.
L 116 – use ‘OS’ instead of ‘oxidative stress’
Reply: We have changed OS’ instead of ‘oxidative stress’
L 118 – correct to Matozzo (Valerio Matozzo) here and in the reference list (#119)
Reply: We have corrected the ms as per the suggestion.
2 Anthropogenic activities and climatic factors….. (influence?)
Reply: We have changed the line as
- Influence of anthropogenic activities and climatic factors on salinity of coastal water
L 156 – change ‘Now days,’ to nowadays
Reply: Yes, we have changed “Now days,’ to nowadays
LL 168,169 ‘where’ or ‘when’?
Re[ply: We have modified the line as Similarly, a lower salinity level is observed under the rate of high precipitation and mixing of runoff water with ocean. And replaced the word where to under.
L 271 ‘Oxidative stress’ change to OS here and in other places after the first mentioning of ‘OS”
Reply: We have changed the Oxidative stress’ change to OS, not only at this place but also throughout the ms. Thank you for the suggestion.
L 272 Why ‘Therefore,’ in this place?
Reply: We have removed the word therefore from this line.
L 275’ catabolism by kinases are’ - catabolism is – correct, please
Reply: We have changed the line as …. energy catabolism by kinases is…..
L 277 – can you give an examples of this difference?
Reply: Yes, we have given the examples of mitogen –activated protein kinase action observed in tilapia (Oreochromis mossambicus) and killifish (Fundulus heteroclitus).
L 281 - '2003) and – put comma
Reply: We put comma as per the suggestion.
L 287 – ‘blood osmotic pressure’ – blood or/and hemolimph?
Reply: Thank you for the suggestion, it could be both, therefore, we have mentioned both.
L 322 -3.2.1. Response of animals to primary and drastic salinization - subdivision impossible (where 3.2.2?)
Reply: 3.2. division was as 3.2. Stress incurred in animals under salinity stress
L 363, 346, 313 – tables, not Table 1-3;
Reply: We have modified this as Tables 1-3
L 279 – why here you separate: (Table 1, Table 2).
Reply: It was a mistake and now been presented as Tables 1-3.
L 693 – unclear: ‘According to their OS physiology is modulated’ (what is modulated?)
Reply: The OS physiology is modulated.
Tables 1-3 Please, provide the references;
Fig 1-2 Please, provide the references;
Reply: We have mentioned in the legend that they were modified after Bal et al. 2021.
Round 2
Reviewer 1 Report
The changes introduced in a hurry by the authors look carelessly treated and continue to reveal lack of knowledge and sensitivity to basic issues, such as the few but fundamental selected ones below.
Abstract and L 67- “modulated by the high rate of evaporation under high temperature, and global warming”.
R: Bad grammar. How do you modulate global warming????
Abstract and L 68- “…..pollutant loads including gases, various dissolved substances and chemicals/ions such as biocrusts, sulfate, sulfuric acid, nitrate, nitric acid, Ca2+, Mg2+, SO4 2− , HCO3− , Na+ and Cl−….”
R: Biocrusts are gases, dissolved substances , or chemicals, or ions???? Likewise for sulfate, sulfuric acid, nitrate, nitric acid; they have chemical formulas, which? Ca2+, Mg2+, SO4 2− , HCO3− , Na+ and Cl−….are what?
L 80- 83- “Salinity is measured as practical salinity unit (PSU), gL -1 , parts per thousand (ppt), parts per million (ppm) etc. Salinity is the saltiness of water, so it is depends on the concentration of salts in soluble form and thus is experimentally measured in terms of salt concentration. For example, it can be measured by quantifying the concentration Cl-…”
R: space/ g L -1; PSU/ grams of what per liter of what? ppt/parts of what per thousand of what? ppm/Parts of what per million of what?; “For example”!!!!/there are reference documents/recommendations which ought to be observed; how did you measure; how do you quantify Cl-? You quantify Cl- and what next; how do you reach salinity values?????
Author Response
The changes introduced in a hurry by the authors look carelessly treated and continue to reveal lack of knowledge and sensitivity to basic issues, such as the few but fundamental selected ones below.
Abstract and L 67- “modulated by the high rate of evaporation under high temperature, and global warming”.
R: Bad grammar. How do you modulate global warming????
Reply: This line is present in Introduction. And we beg apology for the grammatical issues. We have not written to modulate global warming rather we wrote that salinity is modulated by global warming not vice versa. We request the reviewer to look into the revised line as “Rise in salinity in coastal and other water bodies is notably modulated by the rate of evaporation under high temperature, and/or global warming”
So, we never mean to modulate global warming.
Abstract and L 68- “…..pollutant loads including gases, various dissolved substances and chemicals/ions such as biocrusts, sulfate, sulfuric acid, nitrate, nitric acid, Ca2+, Mg2+, SO4 2− , HCO3− , Na+ and Cl−….”
R: Biocrusts are gases, dissolved substances , or chemicals, or ions???? Likewise for sulfate, sulfuric acid, nitrate, nitric acid; they have chemical formulas, which? Ca2+, Mg2+, SO4 2− , HCO3− , Na+ and Cl−….are what?
Reply: we have changed the line as in abstract as
“Besides above, many other anthropogenic, climatic factors, chemicals etc also contribute to the change in salinity in coastal water. Some of them are rain fall, regional warming, precipitation, moisture, thermohaline circulation, gaseous pollutants, dissolved chemicals, wind flow and biocrusts.”
And in introduction as
Many other anthropogenic induced climatic factors, chemicals, ions etc. also modulate the salinity of costal water. Variation in rainfall, regional warming, precipitation rate, thermohaline circulation, biocrusts, pollutant loads including gases, various dissolved substances and chemicals/ions such as sulfate, sulfuric acid, nitrate, nitric acid, Ca2+, Mg2+, SO42−, HCO3−, Na+ and Cl− are few to cite them.
L 80- 83- “Salinity is measured as practical salinity unit (PSU), gL -1 , parts per thousand (ppt), parts per million (ppm) etc. Salinity is the saltiness of water, so it is depends on the concentration of salts in soluble form and thus is experimentally measured in terms of salt concentration. For example, it can be measured by quantifying the concentration Cl-…”
R: space/ g L -1; PSU/ grams of what per liter of what? ppt/parts of what per thousand of what? ppm/Parts of what per million of what?; “For example”!!!!/there are reference documents/recommendations which ought to be observed; how did you measure; how do you quantify Cl-? You quantify Cl- and what next; how do you reach salinity values?????
Reply: ppt/ppm/psu are standard units for the measurement of salinity. It is understood that salinity means grams, mg or ng of salts per little of water. It is general chemistry and therefore not described. Therefore, we had not describe the general fact, however after he comments we have described them in detail.
Cl- is measured biochemically to convert the data into ppt. The reference is Robinson, R.A. The vapour pressure and osmotic equivalence of sea water. F. Mar. Biol. Assoc. U. K. 1954, 33, 449–455. Our paper is not a methodological paper, therefore we did not want to describe in detail how to measure salinity and what is its unit. Therefore, we had just dedicated one para about it. Hope the respected reviewer will understand it.
We have modified the paragraph as “Salinity is measured as practical salinity unit (PSU), gL-1 or parts per thousand (ppt) (i.e. grams of dry weight of salt per kilogram of water), parts per million (ppm i.e. 0.0001 % (w/v) salt in the water) etc. Salinity is the saltiness of water, so it is depends on the concentration of salts in soluble form and thus is experimentally measured in terms of salt concentration. For example, it can be quantified and expressed in terms of the concentration chlorinities of sea water or as molalities (moles of salts per kilogram of water) of sodium chloride solution because both have the same vapour pressure”